# A Nationwide Survey Investigating the Current Status of Genetic Counseling in Newborn Screening in Japan

**DOI:** 10.3390/ijns11040109

**Published:** 2025-11-28

**Authors:** Eri Sakai, Takahiro Yamada, Takashi Hamazaki, Go Tajima, Toshiyuki Seto

**Affiliations:** 1Department of Medical Genetics, Osaka Metropolitan University Graduate School of Medicine, Osaka 545-8585, Japan; sakai.eri@omu.ac.jp; 2Division of Clinical Genetics, Hokkaido University Hospital, Sapporo 060-8648, Japan; 3Department of Pediatrics, Osaka Metropolitan University Graduate School of Medicine, Osaka 545-8585, Japan; 4Division of Neonatal Screening, Research Institute, National Center for Child Health and Development, Tokyo 157-8535, Japan

**Keywords:** newborn screening, neonatal screening, genetic counseling, clinical geneticist, genetic counselor, psychosocial, multidisciplinary care

## Abstract

Following Newborn Screening (NBS), parents receiving positive results experience various psychosocial effects upon learning their child’s genetic information or unexpected findings. These factors warrant careful consideration. The Japanese Medical Association’s Guidelines for Genetic Testing and Diagnosis in Medical Care highlight the importance of genetic counseling (GC) in NBS; however, its current implementation status remains unclear. This study aimed to determine current approaches to GC following positive NBS results in Japan. A questionnaire was conducted with pediatric metabolic specialists responsible for treating individuals who screen positive through NBS results to evaluate GC implementation and their views on its provision. GC was provided at most referral centers for NBS (although not routinely at approximately half of the facilities). In over 70% of cases, GC was performed by a metabolic specialist, regardless of clinical geneticist certification. Furthermore, some metabolic specialists may be reluctant to provide GC due to limited understanding or time constraints. Raising awareness that all parents are eligible for GC, regardless of their child’s diagnosis or health status, is essential. In addition, a GC system incorporating multidisciplinary and multidepartmental collaboration is important for the multifaceted support of patients and families.

## 1. Introduction

Newborn screening (NBS) is a public health program aimed at preventing severe disabilities and reduced life expectancy caused by congenital diseases through early diagnosis; it has been implemented globally since the introduction of the Guthrie test in the 1960s. In Japan, after pilot studies, a publicly funded NBS program was implemented in 1977 as a population-based screening initiative to improve national health, and all newborns were required to undergo the test. Currently, approximately 20 diseases are covered by NBS, including disorders of amino acids, organic acids, and fatty acid metabolism. In addition, an increasing number of municipalities offer optional screening (at the patient’s own expense) for conditions such as spinal muscular atrophy, severe combined immunodeficiency syndrome, and lysosomal diseases [1]. With the widespread implementation of NBS, the diagnosis of rare diseases has become increasingly feasible. However, in recent years, the conditions included in NBS panels have expanded to encompass disorders that may not be fully amenable to effective treatment, along with cases involving false positives and variants of uncertain significance. These characteristics differ substantially from those of classical target conditions and have prompted ongoing debates regarding the appropriateness of screening for such disorders from medical, economic, and psychosocial perspectives [2,3]. This situation suggests the need to reconsider the current support system for patients and their families.

Genetic counseling (GC) is the process of helping people understand and adapt to the medical, psychological and familial implications of genetic contributions to disease [4]. In Japan, 1964 clinical geneticists (approximately 15.8 per million population, as of 2025) and 428 certified genetic counselors (CGCs) (approximately 3.5 per million population, as of 2025) are recognized as genetic professionals who provide GC. Clinical geneticists are designated subspecialists in Japan’s medical specialist system. Certification requires the completion of a primary specialty, followed by three to five years of additional training and successful completion of a qualifying examination. After obtaining certification, they choose whether to hold concurrent appointments in both their original clinical department and the clinical genetics division (and, if so, which is considered their primary affiliation), or to belong solely to the clinical genetics division. Thus, while the clinical genetics division functions as an independent clinical department, it is not uncommon for clinical geneticists to concurrently hold positions in other clinical departments. Japanese certified genetic counselors (non-physicians) are certified by the Japanese Society of Human Genetics and the Japanese Society of Genetic Counseling after completing a designated graduate-level training program and passing a certification examination [5,6,7]. One distinguishing feature of GC in Japan is that only physicians are legally permitted to provide it as a formal medical practice. Although it is preferable for physicians providing GC to be clinical geneticists, certification is not mandatory if they possess adequate knowledge and competencies in GC; however, no objective criteria exist for determining such proficiency. CGCs are non-physicians and their role is classified as medical assistance. Thus they cannot provide GC independently. Therefore, CGCs work in collaboration with clinical geneticists to supplement their workload and provide psychosocial support.

Many of the diseases identified through NBS are hereditary, and professional statements and guidelines recommend GC for individuals with positive results [8,9,10,11]. Although the importance of GC in NBS is acknowledged in the Guidelines for Genetic Tests and Diagnosis in Medical Practice by the Japan Association of Medical Sciences [5], no studies have examined the current status of GC in Japan. This study aimed to assess the current status of GC in the context of NBS in Japan. It provides a foundation for identifying current challenges and necessary improvements, thereby informing the development of an appropriate GC system in anticipation of future NBS expansion.

## 2. Materials and Methods

### 2.1. Survey Methods

This survey was consisted of a web-based questionnaire designed to determine the status of GC implementation at Japanese referral center for NBS as of May 2024. In Japan, patients who screen positive through NBS are referred to the local referral center for NBS in each prefecture for diagnosis and treatment. Therefore, one person from each prefecture was selected for this survey based on a list of physicians designated as “pediatricians who play a central role in NBS programs in each municipality in Japan (metabolic specialist for diagnostic evaluation and treatment),” which is available upon approval from the Japanese Society of Neonatal Screening.

### 2.2. Data Collection Procedures

The survey used an internet response system, Research Electronic Data Capture (REDcap), to transmit and collect data. The content was developed based on input from several physicians and a CGC who are routinely involved in NBS and specialize in pediatrics, obstetrics, gynecology, and clinical genetics. The contents of the questionnaire are summarized below.

Attributes of respondentsCharacteristics of their institutions (e.g., number of genetic professionals)GC implementation status for patients who screen positive through NBS (*): conditions for implementation, people in charge, etc.Necessity and issues of GC

* GC on the implementation of genetic testing (nucleic acid analysis) in infants. The GC performance status was categorized as “routine performance” for all parents who screened positive through NBS, “conditional performance” when conducted under specific conditions (e.g., reproductive intentions or need for parental support), and “not performed” when not carried out. Details of the questionnaire are presented in Appendix A.

### 2.3. Data Analysis

After calculating totals and proportions using descriptive statistics, we analyzed the collaboration between metabolic specialists and genetic professionals. Cases in which a clinical geneticist was involved in GC were categorized as “collaboration with a clinical geneticist,” and those without collaboration were grouped as “no collaboration with a clinical geneticist.” However, if the clinical geneticist involved in GC was also the metabolic specialist responsible for childcare, the case was classified as “no collaboration with a clinical geneticist.” Moreover, we calculated the proportion of CGC involvement in each group. First, a chi-square test was conducted using EZR to assess the presence of collaboration with each genetic professionals, the number of genetic professionals employed, and the presence of a genetic department. The same test was then used to compare CGC involvement between the “collaboration with a clinical geneticist” and “no collaboration with a clinical geneticist” groups. The significance level was set at *p* < 0.05.

## 3. Results

### 3.1. Attribution

All participants responded to the survey. The respondents were affiliated with the following institutions: university hospitals (39/47, 83.0%); other general hospitals (4/47, 8.5%); perinatal medical centers other than university hospitals (3/47, 6.4%); and other institutions (1/47, 2.1%). All respondents were pediatricians, with 44 (94.0%) affiliated with pediatrics departments and 3 (6.0%) with the Division of Clinical Genetics primarily. Fifteen respondents (15/47, 32.0%) were clinical geneticists. All respondents were familiar with GC; over 90% reported routinely treating genetic disorders and having experience referring patients to GC, while 81% stated they had experience in providing GC (Table 1). These findings indicate that most metabolic specialists are pediatricians with extensive experience in genetic medicine while only 32% of them possess certification as clinical geneticists.

### 3.2. Genetic Care System in Referral Center for NBS

Approximately 90% of the centers had a clinical genetics division (41/47, 87.2%). Although 31 centers (66.0%) had at least five clinical geneticists, the most common response indicated that no clinical geneticists were primarily assigned to the Division of Clinical Genetics (21/46, 45.7%). The number of genetic counselors varied across institutions (Appendix A). Although most centers had a genetic department, both the number of personnel and the quality of services varied. In particular, a relatively high proportion of clinical geneticists were assigned not only to genetics departments but also to other clinical departments.

### 3.3. Response to Positive Newborn Screening Cases

#### 3.3.1. Implementation of Genetic Counseling in Genetic Testing for Newborns

Forty-five centers performing genetic testing for NBS-positive newborns were asked about the implementation status of GC. GC was routinely performed in 21 centers (46.7%), conditionally performed in 22 (48.9%) (see Table 2 for details), and not performed in 2 (4.4%). GC was performed in 25 pediatric departments (58.1%) and 18 Divisions of Clinical Genetics (41.9%). The most frequently selected personnel for GC were 31 metabolic specialists (72.1%), of whom 15 (48.4%) were clinical geneticists. Clinical geneticists, who were not metabolic specialists, were selected by the majority, while CGC were selected by less than half of the respondents (multiple responses allowed) (Table 2). As shown above, most referral centers for NBS in Japan provided GC for patients who screen positive through NBS, although it is not routinely implemented in approximately half of them. The results also showed that the majority of GC was provided by metabolic specialists (>70%), regardless of whether they were clinical geneticists. Conversely, over 60% of the facilities had clinical geneticists, other than metabolic specialists, participating in GC.

Respondents were asked about the implementation of GC for individuals who screen positive through NBS. Forty-three of the 45 facilities provided GC. The most frequently selected GC providers were, in descending order, metabolic specialists and clinical geneticists other than metabolic specialists. Among the metabolic specialists selected as GC providers, 16 (51.6%) qualified as clinical geneticists.

#### 3.3.2. Cooperation Between Metabolic Specialists and Genetic Professionals

We examined the status of collaboration between metabolic specialists and genetic professionals in the provision of GC. First, using the data on GC personnel in Table 2, we identified the combinations of physicians selected at each facility and the involvement of CGC (Figure 1a). These combinations were further classified into two groups based on whether clinical geneticists other than metabolic specialists were involved: without collaboration, 14 centers (34.1%), with collaboration, 27 centers (65.9%). In these two groups, CGC involvement was reported in 2 of 14 centers (14.3%) and 13 of 27 centers (48.1%), respectively. Data analysis showed that the proportion of CGC involvement was significantly higher (*p* = 0.044) in the group collaborating with a clinical geneticist (Figure 1b). In the group of no collaboration with clinical geneticist, seven metabolic specialists were also clinical geneticists. To evaluate the potential influence of this factor on the results, we further compared two subgroups: metabolic specialists who were clinical geneticists vs. those who were not. No differences were observed (the first and second bars in Figure 1a). Therefore, we concluded that whether the metabolic specialist was a clinical geneticist did not affect the involvement of CGC. These subgroups were combined in the primary analysis (Figure 1b). The results indicate that both clinical geneticists and the CGC tend to be involved when metabolic specialists collaborate with other clinical geneticists. Collaboration between metabolic specialists and genetic professionals in NBS did not always occur, even when the number of genetic professionals at a facility increased.

### 3.4. Opinions of Metabolic Specialists Regarding GC

Respondents were asked to provide their opinions regarding GC for individuals who tested positive through NBS. All 46 respondents except one reported that “GC is useful for follow-up of the child (initiation and continuation of treatment)” and “useful for follow-up of the parents (consideration of future pregnancies and health care)” while one respondent stated “I am not sure about the usefulness of GC. The most frequently cited expectations for GC for individuals who screen positive through NBS were “consultation regarding future pregnancies (89.4%),” “providing and organizing genetic information (85.1%),” and “listening to patient and family anxiety and conflicts (78.7%)” (Appendix A). Conversely, 21 respondents indicated that they “sometimes do not perform GC on individuals who screen positive through NBS”. The most common reasons cited were “the problem can be addressed through regular pediatric care” (13/21, 61.9%) and “the child was in good condition and did not require medical intervention” (8/21, 38.1%). Approximately 76.2% (16/21) of respondents cited one of these two reasons (Figure 2). These findings suggest that some metabolic specialists perceived an overlap between the roles of GC and pediatric care, viewing GC as an extension of pediatric care in certain cases.

## 4. Discussion

This study clarified the current approaches to GC for NBS-positive cases across Japan and examined metabolic specialists’ perceptions regarding the provision of GC. Based on these findings, we discuss the current challenges of GC for parents of NBS-positive infants and the future development of GC provision.

### 4.1. Current Status of GC in Japan—Genetic Consultation by Metabolic Specialists

This survey revealed that metabolic specialists were more likely than clinical geneticists or CGCs to provide GC within the Japanese NBS system (Table 2). This tendency may be attributed to the historical and cultural context in which the NBS developed in Japan. In the 1970s, when NBS was first launched, there were no specialists in Japan trained in GC, such as clinical geneticists or CGC, and GC—then referred to as “genetic consultation”—was provided by physicians as part of routine patient care. In the context of NBS, metabolic specialists had already established trusting relationships with patients and their families and possessed in-depth knowledge of the natural history and genetics of relevant disorders. Therefore, it was considered standard practice at that time for metabolic specialists to provide GC as part of pediatric care. In the 21st century, training programs for genetic professionals and the development of a genetic medicine system have progressed rapidly in Japan. However, the practice of metabolic specialists providing GC as an extension of general medical care remains deeply embedded in the NBS system.

### 4.2. Metabolic Specialists Tend to Understand Only the Limited Role of GC

While most respondents acknowledged the usefulness of GC for individuals who screen positive through NBS, some perceived its role as overlapping with that of standard medical care (Figure 2). Individuals who screen positive through NBS may face a range of psychosocial challenges, including anxiety and confusion due to unexpected results, decisions about future reproduction, carrier status, and concerns regarding the genetics of blood relatives. These concerns are difficult to address within the constraints of time-limited pediatric care, and affected individuals and their families often experience ongoing conflict [12]. Previous studies have demonstrated that GC benefits for individuals who tested positive through NBS include reduced parental anxiety, more efficient diagnosis of the child, improved parental understanding of results, provision of future reproductive options, enhanced patient-family communication, and diagnosis of relatives [12,13,14,15,16,17]. In other words, GC enables parents to “accept the results and adapt to the child’s diagnosis and treatment,” or to use the information provided to “prepare for future challenges,” even if the child is healthy, highlighting a quality of care that differs from conventional medical practice. Thus, these perceptions among metabolic specialists may be inappropriate, suggesting a limited understanding of the scope of GC. It is also important to note that the challenges faced by pediatric patients and their families—and their severity—are not always predictable based on disease severity [18,19,20]. Therefore, metabolic specialists should consider GC for all individuals who screen positive through NBS.

### 4.3. Involvement of Clinical Geneticists in GC Results in Increased Involvement of the CGC

The results of the data analysis showed that the proportion of CGCs involved in GC was significantly higher in facilities where metabolic specialists collaborated with clinical geneticists (Figure 1). As mentioned previously, in Japan, only physicians are legally permitted to provide GC as a medical practice; however, it is difficult for physicians alone to manage all tasks, and a cooperative system between clinical geneticists and CGCs is considered ideal. Given the current situation, the analysis suggests that the involvement of clinical geneticists may facilitate support from CGCs and promote multidisciplinary collaboration. A GC system with a multidisciplinary team is extremely important for NBS and requires flexible support for patients with positive results.

However, several factors hinder collaboration among genetics professionals. The first factor involves the perceptions of metabolic specialists. The previous results indicate a lack of understanding of the significance of GC. It is necessary to conduct educational campaigns targeting metabolic specialists and other pediatricians. Second, there is a shortage of genetics professionals. The number of clinical geneticists in Japan is not low compared to Western countries when adjusted for population size [6,21]. However, unlike the United States, where clinical genetics is recognized as a primary specialty (American Board of Medical Genetics and Genomics), in Japan, it is a subspecialty pursued after obtaining certification in a primary field. Therefore, clinical geneticists are often required to practice concurrently in both the primary specialty and the clinical genetics division, which can easily limit their efforts. In addition, the shortage of CGC remains a challenge due to persistent understaffing [22]. Finally, there is an institutional challenge. The placement of genetic professionals may be more likely to be prioritized in areas where GC is institutionally mandated (e.g., cancer genome medicine and prenatal testing). Therefore, it is necessary to improve the NBS implementation system to mandate the development of GC systems. Furthermore, the current GC fee structure in Japan adds charges only at the time of disclosing limited test results; however, the system should allow reimbursement at the time of GC implementation involving genetic professionals.

### 4.4. Other: Issues Related to GC of NBS

In addition, some issues remain concerning GC in NBS. First, obstetricians, gynecologists, and midwives need further education on NBS. Previous studies have indicated that improving parental understanding may mitigate some of the psychosocial challenges. Moreover, evidence suggests that parents tend to prefer prenatal education [23,24,25]. Thus, these health care providers should understand the prevalence of congenital diseases and the importance of NBS. In recent years, opportunities to provide parental testing information early in pregnancy have increased, making it feasible to raise public awareness of NBS during these encounters [26,27]. Second, information about GC and NBS must be disseminated to the public. A survey of parents of children who screen positive through NBS reported that many were unaware of the availability of GCs [28]. Third, GC must be adapted over time [29]. GC opportunities should align with parents’ life stages and the child’s developmental milestones. Finally, regional disparities in GC access persist. Clinical geneticists and CGC tend to be concentrated in urban centers, creating a lack of access for people in rural areas [22]. There are validated cases and recommended guidelines for remote GC in NBS-positive patients overseas, which should be further discussed in Japan [11,30].

### 4.5. Limitations and Perspectives of the Study

The survey targeted Japanese metabolic specialist. A survey targeting professionals involved in NBS (pediatricians other than metabolic specialists, obstetricians/gynecologists, midwives, genetic counselors) is required to gain a more comprehensive understanding of the current situation. The survey did not assess the respondents’ level of knowledge regarding GC; therefore, we were unable to determine whether they possessed sufficient knowledge to serve as appropriate providers of GC. In addition, this survey does not consider differences in the number of births by prefecture and fails to reflect differences in the situation at each facility. The survey focused on the current status of population-based screening for all births; however, a separate survey is needed to assess the status of optional screening conducted by individual municipalities.

## 5. Conclusions

In Japan, GC for individuals who screen positive through NBS is currently provided by metabolic specialists and/or clinical geneticists, with CGC detected in fewer than half of the cases. Furthermore, it has been revealed that some metabolic specialists may be reluctant to provide GC due to an insufficient understanding. Improving the GC system requires enhanced collaboration between metabolic specialists and genetic professionals to provide comprehensive support for individuals who screen positive through NBS.

## Figures and Tables

**Figure 1 IJNS-11-00109-f001:**
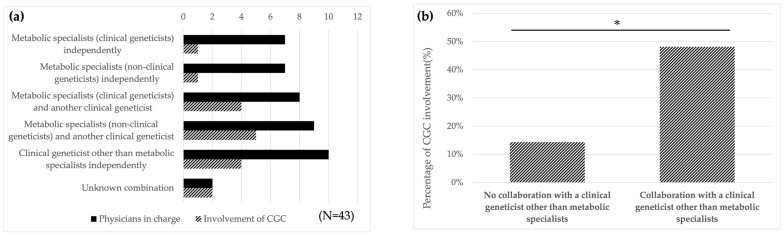
Collaboration with clinical geneticists other than metabolic specialists in GC results in increased involvement of CGC. (**a**) For GC provider in Table 2, the combination of physicians in charge selected at each facility (black) and the number of facilities in which a CGC is involved (shaded) are indicated. In 14 centers, GC was conducted solely by metabolic specialists (the first and second black bars), whereas in 27 centers, it was carried out in collaboration with clinical geneticists (the third to fifth black bars). Note that metabolic specialists are distinguished as either clinical geneticists or non-clinical geneticists. (**b**) Patterns in Figure 1a were classified into two groups according to the involvement of clinical geneticists other than metabolic specialists. The proportion of CGC involvement in GC was 14.3% in hospitals with no collaboration with a clinical geneticist other than metabolic specialists and 48.1% in hospitals with collaboration with a clinical geneticist other than metabolic specialists. The results of the chi-square test showed a difference in the proportion of involvement of CGC. (* *p* < 0.05).

**Figure 2 IJNS-11-00109-f002:**
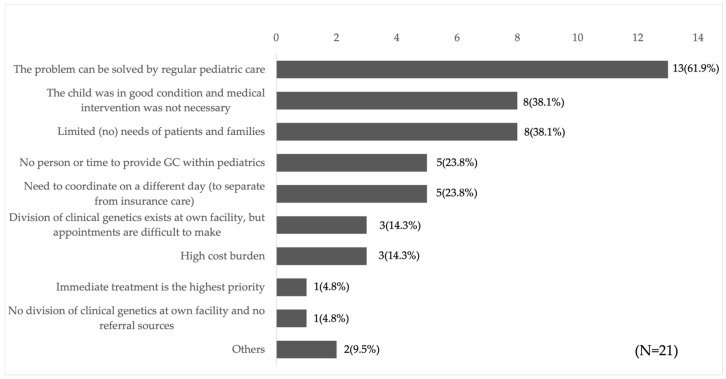
Reasons why GC is not implemented. Twenty-one respondents who answered that they “sometimes do not perform GC on individuals who screen positive through NBS” indicated the reason why they do not.

**Table 1 IJNS-11-00109-t001:** Attribution.

Questions/Answers	*n*	(%)
About the Respondent (*N* = 47)		
Q1. Area of expertise		
Pediatrics	47	100%
Q2. Qualifications of clinical geneticists		
Qualified	15	31.9%
Not Qualified	32	68.1%
Q3. Main department		
Pediatrics	44 ^1^	93.6%
Division of clinical genetics	3 ^2^	6.4%
Respondent’s experience/knowledge of GC (*N* = 47)		
Q4. Do you routinely treat genetic disorders?		
Yes	43	91.5%
No	4	8.5%
Q5. Do you know about GC?		
Yes	47	100%
Q6. Do you have a referral source if you think GC is needed?		
Yes	46	97.9%
No	1	2.1%
Q7. Do you have experience referring patients to GC?		
Yes	45	95.7%
No	2	4.3%
Q8. Do you have experience in providing GC?		
Yes	38	80.9%
No	9	19.1%

^1^ 4 respondents also belong to the Division of Clinical Genetics. ^2^ 2 respondents also belong to Pediatrics.

**Table 2 IJNS-11-00109-t002:** Implementation of GC in Child Genetic Testing.

Questions/Answers	*n*	(%)
Q1. GC Implementation (*N* = 45)		
Routine performance	21	46.7%
Conditional performance (continued in Q2)	22	48.9%
Not performed	2	4.4%
Q2. Conditions for conducting GC (*N* = 22)		
Future reproductive consultation	17	77.3%
Interpretation of genetic test results	15	68.2%
Suspicious family history or at-risk individuals	14	63.6%
Parents need care	13	59.1%
Consideration of prenatal/preimplantation testing	13	59.1%
Need assistance in explaining the disease overview	12	54.5%
Disease-specific considerations	3	13.6%
Coordination of in-facility and out-of-facility care	2	9.1%
Q3. Place of implementation (*N* = 43)		
Pediatrics	25	58.1%
Division of clinical genetics	18	41.9%
Q4. GC provider (*N* = 43) Multiple choice		
Metabolic specialists	31	72.1%
Clinical geneticists other than metabolic specialists	27	62.8%
Non-clinical geneticists other than metabolic specialists	2	4.7%
CGC (Not by oneself)	17	39.5%

## Data Availability

The original contributions presented in this study are included in the article/Appendix A. Further inquiries can be directed to the corresponding author.

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
