# Peer review of "A Nationwide Survey Investigating the Current Status of Genetic Counseling in Newborn Screening in Japan"

_2409-515X, 2025, doi:10.3390/ijns11040109_

Round 1

Reviewer 1 Report

Comments and Suggestions for Authors

This manuscript reports the examination of health care professionals in Japan who see patients identified through newborn screening and provide genetic counseling to their parents.  The authors state that the Japanese Medical Association guidelines highlight the importance of genetic counseling in newborn screening but the implementation of this guideline has been unknown.  The authors conducted a questionnaire survey with pediatric metabolic specialists who treat individuals who screen positive through NBS to explore the provision of genetic counseling and the providers’ views on genetic counseling.  While the underlying message based on the data collected in this project is important, there are several areas that need clarification.

Page 2, line 46:  Information regarding the reported inclusion of conditions in the Japanese newborn screening panel that are not treatable and that variants of uncertain significance are being found is not expanded in the manuscript.  What is the significance of this information?   Is this adding to the need for genetic counseling beyond that usually provided for standard newborn screening conditions?  If so, this should be stated.  This information does not seem pertinent for the remainder of the manuscript.

Page 3, lines 92-96: It is unclear what the bullet points are referring to.  Do they represent a summary of the content in the survey?  If so, then this should be stated. A copy of the survey questions in the Supplemental materials would be helpful. 

Page 3, lines 101-105 and Figure 1b: “Cases in which a clinical geneticist or CGC was involved in GC were categorized as “collaboration,” and those without collaboration were grouped as “no collaboration.” However, if the clinical geneticist involved in GC was also the metabolic specialist responsible for childcare, the case was classified as “no collaboration.”” It is not clear what this means.  If a physician providing care for a child with a metabolic disorder identified through newborn screening (I would consider this a metabolic specialist) is also a clinical geneticist and a CGC is involved with the case, why would this not be considered a “collaboration”? Reconsidering how these cases are categorized may influence the data analysis. 

Table I, Question 2:  68.1% of the respondents were “not qualified”.  Does this mean that they trained in genetics but did not pass the qualifying examination?  As this is an international journal and in some countries “not qualified” means that a health care provider is not qualified to practice medicine, this needs further explanation somewhere.  If they are not qualified, how are they able to still practice medicine? 

Page 3, line 120:  Regarding providers personal experience in genetic counseling.  Did this question asked if the providers themselves had experience providing genetic counseling?  If so, this should be stated as this question is not clear.  It could also mean that they have personally received genetic counseling. 

Page 3, line 121-122:  “These findings indicate that metabolic specialists, who are also pediatricians, possess extensive experience in genetic medicine.”  From the data presented it seems to indicate that although they have experience in genetic medicine they are not qualified as clinical geneticists.  This section needs further explanation and clarification. 

Page 4, Section 3.2:  It would be helpful to describe how “clinical genetics divisions” are typically structured in Japan.  In the U.S., many are under a primary department such as Pediatrics or Internal Medicine, but some may be under a Department of Genetics.  Are the authors able to tell what the structure is of the Departments/Divisions of their respondents?  How was the quality of services determined?  That a Division of Clinical Genetics was also a department is very confusing and should be explained.  Do most clinical geneticists work in departments that don’t have a Division of Clinical Genetics?  Do they have primary appointments in other departments? 

Page 5, Section 3.3.1:  What is meant by “conditionally performed” genetic counseling here?  Do the questions here refer to all cases of childhood genetic testing and not just those related to newborn screening?  What is the difference between routine and conditional performance? This is confusing. 

Page 5, Section 3.3.2:  What do the 14 versus 27 centers indicate?  It is unclear which ones have involvement of clinical geneticists. This requires clarification.

Page 8, Section 4.1:  The statement that “metabolic specialists are more likely to provide genetic counseling… “ should be clarified to reflect the data in Table 2 as from the statement it leaves the reader wondering “more likely than what?”.  Metabolic specialists who are not trained in genetics are providing more genetic counseling in Japan as compared to qualified clinical geneticists (at least I think this is what is meant), but this only reflects that provided for childhood genetic testing in general and is not specific for genetic counseling provided for families whose infants have positive newborn screening. 

Page 9, Section 4.4:  While Ob-Gyns and nurse midwives likely do need further education in genetic counseling, I do not see data provided regarding this.  This may be the authors’ opinion but not substantiated by their data and best deleted or a reference provided. 

Comments on the Quality of English Language

There are some minor grammatical errors but this does not detract from the overall manuscript.  

Author Response

We sincerely thank the reviewers for their valuable and constructive comments. We have carefully revised the manuscript according to the suggestions, and the responses are described below in blue point by point. All modifications have been highlighted in the revised manuscript using the track changes function for ease of reference.

Response to Reviewer1 and Editor

Comment1:

Page 2, line 46:  Information regarding the reported inclusion of conditions in the Japanese newborn screening panel that are not treatable and that variants of uncertain significance are being found is not expanded in the manuscript.  What is the significance of this information?   Is this adding to the need for genetic counseling beyond that usually provided for standard newborn screening conditions?  If so, this should be stated.  This information does not seem pertinent for the remainder of the manuscript.

Response:

We appreciate your insightful comment. As you point out, it is unclear if this information about the expansion of NBS pertinent for the remainder of the manuscript. This description was included to highlight the importance of reexamining the support system for patients and their families such as GC, where the expansion of NBS presents various challenges to patients. To make this intention clearer, we have added explanatory text accordingly.

Revised text in the manuscript:

“This situation suggests the need to reconsider the current support system for patients and their families. (page 2, lines 51-53)”

Comment2:

Page 3, lines 92-96: It is unclear what the bullet points are referring to.  Do they represent a summary of the content in the survey?  If so, then this should be stated. A copy of the survey questions in the Supplemental materials would be helpful. 

Response:

We sincerely appreciate for this important comment. As you mentioned, this bullet points represent a summary of the content of the questionnaire. We have added the explanation below. As the questions and response options are already provided in the main text, we have omitted them here to avoid redundancy.

Revised text in the manuscript:

“The contents of the questionnaire are summarized below. (page 2, lines 100)”

Comment3:

Page 3, lines 101-105 and Figure 1b: “Cases in which a clinical geneticist or CGC was involved in GC were categorized as “collaboration,” and those without collaboration were grouped as “no collaboration.” However, if the clinical geneticist involved in GC was also the metabolic specialist responsible for childcare, the case was classified as “no collaboration.”” It is not clear what this means.  If a physician providing care for a child with a metabolic disorder identified through newborn screening (I would consider this a metabolic specialist) is also a clinical geneticist and a CGC is involved with the case, why would this not be considered a “collaboration”? Reconsidering how these cases are categorized may influence the data analysis. 

Response:

We appreciate the valuable and thoughtful comment. As you pointed out, it was unclear what was classified as “collaboration”. Actually, we treated the involvement of clinical geneticists and that of CGC as separate variables. To clarify the distinguish, we added new explanations regarding “Collaboration with a clinical geneticists” and “Involvement of CGC”.

Revised text in the manuscript:

“Cases in which a clinical geneticist was involved in GC were categorized as “collaboration with a clinical geneticist,” and those without collaboration were grouped as “no collaboration with a clinical geneticist.” … Moreover, we calculated the proportion of CGC involvement in each group. (page 3, lines 114-120)”

“These combinations were further classified into two groups … (page 6, lines 196)”

“Figure1. Collaboration with clinical geneticists other than metabolic specialists in GC results in increased involvement of CGC. (page 7, Figure1 title)”

Comment4:

Table I, Question 2:  68.1% of the respondents were “not qualified”.  Does this mean that they trained in genetics but did not pass the qualifying examination?  As this is an international journal and in some countries “not qualified” means that a health care provider is not qualified to practice medicine, this needs further explanation somewhere.  If they are not qualified, how are they able to still practice medicine? 

Response:

We appreciate the helpful comment. In Japan, it is common that the GC is provided by both clinical geneticists and CGC (as described in page 2, lines 77). On the other hand, physician who are not clinical geneticists are also permitted to provide GC independently if they possess sufficient knowledge and skills in GC. To improve manuscript, we added the following clarification to explain it.

Revised text in the manuscript:

“physicians are not necessarily required to be clinical geneticists, provided that they possess adequate knowledge and competence in GC. (page 2, lines 73-74)”

Comment5:

Page 3, line 120:  Regarding providers personal experience in genetic counseling.  Did this question asked if the providers themselves had experience providing genetic counseling?  If so, this should be stated as this question is not clear.  It could also mean that they have personally received genetic counseling. 

Response:

Thank you for your valuable comment. As you pointed out, the question “Do you have personal experience in GC?” was a misleading expression. To clarify that we asked whether the respondents have experience in providing GC, we have changed it to the expression “Do you have experience in providing GC?”.

Revised text in the manuscript:

“Do you have experience in providing GC? (page 4, Table1)”

“while 81% stated they had experience in providing GC (page 4, lines 138-141)”

Comment6:

Page 3, line 121-122: “These findings indicate that metabolic specialists, who are also pediatricians, possess extensive experience in genetic medicine.”  From the data presented it seems to indicate that although they have experience in genetic medicine they are not qualified as clinical geneticists.  This section needs further explanation and clarification. 

Response:

Thank you for your constructive suggestion. As you pointed out, only 32% of metabolic specialists possess certification as clinical geneticists although metabolic specialists have experience in genetic medicine, which should be explicitly stated as a characteristic of the data. We have revised it to improve the manuscript.

Revised text in the manuscript:

“These findings indicate that most metabolic specialists are pediatricians with extensive experience in genetic medicine while only 32% of them possess certification as clinical geneticists. (page 4, lines 138-141)”

Comment7:

Page 4, Section 3.2:  It would be helpful to describe how “clinical genetics divisions” are typically structured in Japan.  In the U.S., many are under a primary department such as Pediatrics or Internal Medicine, but some may be under a Department of Genetics.  Are the authors able to tell what the structure is of the Departments/Divisions of their respondents?  How was the quality of services determined?  That a Division of Clinical Genetics was also a department is very confusing and should be explained.  Do most clinical geneticists work in departments that don’t have a Division of Clinical Genetics?  Do they have primary appointments in other departments? 

Response:

I appreciate your constructive suggestion. As you pointed out, the explanation of how genetic medicine is structured in Japan and how clinical geneticists work was insufficient. We added the additional explanation below to improve the manuscript.

Revised text in the manuscript:

“After obtaining certification, they choose whether to hold concurrent appointments in both their original clinical department and the clinical genetics division (and, if so, which is considered their primary affiliation), or to belong solely to the clinical genetics division. Thus, while the clinical genetics division functions as an independent clinical department, it is not uncommon for clinical geneticists to concurrently hold positions in other clinical departments. (page 2, lines 62-68)”

Comment8:

Page 5, Section 3.3.1:  What is meant by “conditionally performed” genetic counseling here?  Do the questions here refer to all cases of childhood genetic testing and not just those related to newborn screening?  What is the difference between routine and conditional performance? This is confusing. 

Response:

We appreciate your valuable comment. As you pointed out, the difference between “routine performance” and “conditionally performed” was unclear. It was categorized as “routine performance” for all parents who screened positive through NBS, and “conditional performance” when conducted under specific conditions (specific conditions are listed in the table2 page 6). Moreover, it was not clear that this survey inquired about the GC for parents who screened positive through NBS, which could cause confusion for readers. To improve the manuscript, we added additional explanations below.

Revised text in the manuscript:

“The GC performance status was categorized as “routine performance” for all parents who screened positive through NBS, “conditional performance” when conducted under specific conditions, and “not performed” when not carried out.” (page 3, lines 107-110)”

“Forty-five centers performing genetic testing for NBS-positive newborns were asked about the implementation status of GC. (page 5, lines 167)”

Comment9:

Page 5, Section 3.3.2:  What do the 14 versus 27 centers indicate?  It is unclear which ones have involvement of clinical geneticists. This requires clarification.

Response:

We appreciate your important comment. As you pointed out, it was unclear what the 14 and 27 centers referred to in the sentence. It describes that GC were conducted independently by metabolic specialists in14 centers, whereas carried out in collaboration with clinical geneticists in 27 centers. To improve the manuscript, we revised the explanations below.

Revised text in the manuscript:

“These combinations were further classified into two groups based on whether clinical geneticists other than metabolic specialists were involved: without collaboration, 14 centers (34.1%), with collaboration, 27 centers (65.9%). (page 6, lines 196)”

“In 14 centers, GC was conducted solely by metabolic specialists (the first and second black bars), whereas in 27 centers, it was carried out in collaboration with clinical geneticists (the third to fifth black bars). (page 7, lines 214-216)”

Comment10:

Page 8, Section 4.1:  The statement that “metabolic specialists are more likely to provide genetic counseling… “should be clarified to reflect the data in Table 2 as from the statement it leaves the reader wondering “more likely than what?”.  Metabolic specialists who are not trained in genetics are providing more genetic counseling in Japan as compared to qualified clinical geneticists (at least I think this is what is meant), but this only reflects that provided for childhood genetic testing in general and is not specific for genetic counseling provided for families whose infants have positive newborn screening. 

Response:

We appreciate your valuable comment. As you pointed out, ”metabolic specialists are more likely to provide genetic counseling…” was unclear what the data compared to. In this sentence, we compared between metabolic specialists and clinical geneticists other than metabolic specialists or CGC. To improve the manuscript, we added the explanation below. As a side note, as described in comment 8, this survey inquired about the GC for parents who screened positive through NBS.

Revised text in the manuscript:

“This survey revealed that metabolic specialists were more likely than clinical geneticists or CGCs to provide GC within the Japanese NBS system. (page 8, lines 254-255)”

Comment11:

Page 9, Section 4.4:  While Ob-Gyns and nurse midwives likely do need further education in genetic counseling, I do not see data provided regarding this.  This may be the authors’ opinion but not substantiated by their data and best deleted or a reference provided.

Response:

We appreciate the valuable suggestion. As you pointed out, the evidence supporting the claim “These health care providers should understand the prevalence of congenital diseases and the importance of NBS”. We revised the text and updated the discussion to reflect evidence from the literature.

Revised text in the manuscript:

“Previous studies have indicated that improving parental understanding may mitigate some of the psychosocial challenges. Moreover, evidence suggests that parents tend to prefer prenatal education [2]. (page 10, lines 324-327)”

Response to Reviewer 1 and Editor

As you pointed out, we have included the approval date in the Institutional Review Board Statement(page11, line371-372). The English was previously reviewed by a native speaker (the certificate is attached), and in this revision we have clarified points that were previously unclear. However, we would greatly appreciate any further suggestions for improvement, should you consider additional refinements necessary.

Reviewer 2 Report

Comments and Suggestions for Authors

This is a very interesting beginning approach to examining genetic counseling and other educational needs of families and how best to serve them.  The authors detail the different attitudes and uptake by various physicians of genetic counseling services in the NBS sphere.

Areas that could help readers:

-Help us understand the difference between clinical geneticist and metabolic specialists early on.  Japan has its own system of physician training, especially as it relates to genetics, which the authors do touch on, but a clear explanation of training and roles early on would add to the clarity and understanding.

-Make it very clear if this is discussing genetic counseling before or after the diagnosis is established; that was a bit unclear to me.

Author Response

We sincerely thank the reviewers for their valuable and constructive comments. We have carefully revised the manuscript according to the suggestions, and the responses are described below in blue point by point. All modifications have been highlighted in the revised manuscript using the track changes function for ease of reference.

Response to Reviewer2

Comment1:

Help us understand the difference between clinical geneticist and metabolic specialists early on. Japan has its own system of physician training, especially as it relates to genetics, which the authors do touch on, but a clear explanation of training and roles early on would add to the clarity and understanding.

Response:

We appreciate your thoughtful comment. As you pointed out, we didn’t provide an adequate explanation of structure of Japanese physician training and roles, which was also pointed out in Comment7 by reviewer1. To imply the manuscript, we added the explanation.

Revised text in the manuscript:

“After obtaining certification, they choose whether to hold concurrent appointments in both their original clinical department and the clinical genetics division (and, if so, which is considered their primary affiliation), or to belong solely to the clinical genetics division. Thus, while the clinical genetics division functions as an independent clinical department, it is not uncommon for clinical geneticists to concurrently hold positions in other clinical departments. (page 2, lines 62-68)”

Comment2:

Make it very clear if this is discussing genetic counseling before or after the diagnosis is established; that was a bit unclear to me.

Response:

Thank you for your valuable comment. As you pointed out, we believe that clearly distinguishing the timing of GC is important. On the other hand, the timing of GC was not strictly defined as before or after diagnosis; it was set as “GC on the implementation of genetic testing (nucleic acid analysis) in infants.” This is because, in NBS, we consider that the time to diagnosis is limited, and the timing of GC may easily vary before or after diagnosis.
